# Comparison of national and international sedentary behaviour and physical activity guidelines for older adults: A systematic review and quality appraisal with AGREE II

Amy Huang [1], Ellen Wang[2,3], Stephanie Sanger[4‡], Alexandra Papaioannou[1,5‡], Isabel B. Rodrigues [1,6] *

1 Department of Medicine, McMaster University, Hamilton, ON, Canada, 2 Department of Physical Therapy, University of British Columbia, Vancouver, BC, Canada, 3 Arthritis Research Canada, Vancouver, BC, Canada, 4 Department of Health Research Methods, Evidence, and Impact, McMaster University, Hamilton, ON, Canada, 5 Health Sciences Library, McMaster University, Hamilton, ON, Canada, 6 Knowledge Translation Program, St. Michael's Hospital, Unity Health Toronto, Toronto, ON, Canada

☯ These authors contributed equally to this work.
‡ These authors also contributed equally to this work.
* rodrigib@mcmaster.ca

**Data Availability Statement:** All relevant data are within the paper and Supporting Information files.

## Abstract

Most older adults 65 years and older accumulate over 8.5 hours/day of sedentary time, which is associated with increased risk of metabolic syndromes and falls. The impact of increased sedentary time in older adults has prompted development of sedentary behaviour guidelines. The purpose of our review was to compare national and international sedentary behaviour and physical activity guidelines for older adults and appraise the quality of guidelines using AGREE II. We conducted our search in Medline, Embase, Global Health, Web of Science, CINAHL, and relevant grey literature. We included the most recent guidelines for older adults written in English. We identified 18 national and international guidelines; ten of the 18 guidelines included sedentary behaviour recommendations while all 18 included physical activity recommendations for older adults. The ten sedentary behaviour guidelines were developed using cohort studies, knowledge users' opinions, systematic reviews, or other guidelines while the physical activity guidelines were developed using randomized controlled trials, systematic reviews, meta-analysis, and overview of reviews. The definition of sedentary behaviour and the recommendations were inconsistent between the guidelines and were based on very low to low quality and certainty of evidence. All guidelines provided consistent recommendations for aerobic and resistance training; the recommendations were developed using moderate to high quality and certainty of evidence. Only eight physical activity guidelines provided recommendations for balance training and six on flexibility training; the balance training recommendations were consistent between guidelines and based on moderate quality evidence. Further work is needed to develop evidenced-based sedentary behaviour recommendations and flexibility training recommendations for older adults.

**Funding:** We received financial support from the Hamilton Health Sciences New Investigators Fund, AGE-WELL-McMaster Institute for Research on Aging (MIRA) Award. The funders had no role in study design, data collection and analysis, decision to publish, or preparation of the manuscript.

**Competing interests:** The authors have declared that no competing interests exist.

## Introduction

Large amounts of sedentary time can be detrimental to health, particularly for those who do not engage in adequate amounts of moderate to vigorous physical activity [1,2]. Sedentary behaviour is defined as "any waking behaviour characterized by an energy expenditure of 1.5 metabolic equivalents (METs) or lower while sitting, reclining, or lying", while sedentary time is measured by "the amount of time spent in these positions" [3]. Two thirds of older adults accumulate over 8.5 hours of sedentary behaviour per day [4]. Prolonged time spent in sedentary behaviours are associated with an increased risk of all-cause mortality, incident cardiovascular disease, and type two diabetes [5]. With the projected increase in the older adult population and corresponding number of individuals engaging in sedentary behaviours, there is pressing need for research that aims to disrupt patterns of sedentary time and integrate movement into daily life.

Older adults have become the most sedentary and least physically active age group [6,7]. There is high certainty evidence indicating a relationship between sedentary behaviour and metabolic syndrome, waist circumference, and obesity [7]. Further, prolonged sedentary behaviours are associated with detrimental effects on bone health and a corresponding increase in the incidence of sarcopenia, fractures, and falls [8]. Thus, interventions aimed at modifying sedentary behaviour and time are essential to minimize the associated negative health outcomes for older adults. Several clinical practice guidelines from around the world provide recommendations on physical activity, exercise, and sedentary time [9–12]; however, there is limited evidence on effective interventions to decrease sedentary time and behaviour among older adults and if such interventions can improve health-related outcomes [13]. There is also insufficient evidence to make recommendations on the frequency and duration of breaks in sedentary behaviour, as well as very limited evidence to set quantified recommendations on sedentary behaviours [14].

In the past decade, numerous countries have reviewed or updated their sedentary behaviour guidelines for older adults, with a trend toward more evidence-based guidelines [9–12]. A synthesis of international sedentary behaviour guidelines can help reveal more information about interventions that may be used to decrease sedentary behaviours, time spent engaging in these behaviours, and methods to decrease total sedentary time in older adults. In addition, most interventions have attempted to decrease sedentary time by increasing physical activity levels with the assumption that sedentary time would be reallocated to physical activity [13]. Therefore, reviewing physical activity guidelines can also provide insight into effective interventions that target sedentary behaviour since most sedentary behaviour guidelines are part of the physical activity guidelines [9–12]. Lastly, comparing sedentary behaviour and physical activity guidelines can provide insight into best practices, avoid duplication, and identify knowledge gaps. The purpose of this systematic review was to separately survey available sedentary behaviour and physical activity guidelines from around the world to compare recommendations and critically analyze the methodology through which the guidelines were developed.

## Methods

### Eligibility criteria

We included physical activity and sedentary behaviour guidelines that were written in English, contained recommendations for older adults, and were the most recent version of the guidelines. Guidelines were identified if the term "guideline" or "clinical practice guideline" was in the title. There were no restrictions on sex, ethnicity, or setting.

## Search strategy/clinical practice guideline identification

A librarian (SS) at McMaster University and experienced in conducting review searches developed a literature search (S1 Table). The literature search was conducted in Ovid Medline, Ovid Embase, Ovid Global Health, and grey literature (e.g., TRIP and Google) on October 20th, 2022, and last updated on October 31st, 2022. The search strategy was developed using a combination of Medical Subject Headings and keywords. We used Agency for Healthcare Research and Quality (https://www.ahrq.gov/gam/index.html) to identify additional guidelines in early December 2022. On August 22nd, 2023, we expanded our search to include Web of Science, and CINAHL. We also searched for additional guidelines through Canada's Drug and health Technology Agency Grey Matters Database (https://www.cadth.ca/) and the Scottish Intercollegiate Guideline Network (https://www.sign.ac.uk/). A second librarian the University of British Columbia reviewed the search strategy.

## Selection process of relevant guidelines

Three independent reviewers (AH, EW, IBR) screened the titles and abstracts to identify relevant guidelines. Data was extracted through Covidence (Veritas Health Innovation, Melbourne, Australia) by at least two independent reviewers (AH, EW). We extracted the following information: the name of the guideline, publication year, country of origin, type stakeholder involved in developing the guideline, stakeholder's conflict of interest, type of evidence used to develop the recommendations and certainty of evidence, terminology, and funding source. We categorized the recommendations using frequency, intensity, time/duration, and type of activity. A third independent reviewer reviewed the extracted information in Covidence and met with the other reviewers if there was a conflict. Conflicts were discussed until a compromised was reached. If there was missing information, we did not contact the authors as the guidelines should be accessible to all individuals.

## Quality assessment

We utilized the Appraisal of Guidelines, Research and Evaluation II (AGREE II) instrument to evaluate the methodological quality of the guidelines (https://www.agreetrust.org) [15]. AGREE II is a validated and reliable tool to evaluate guidelines [15]. Each clinical practice guideline was scored on 23 items of six domains [15]. Two reviewers (AH and EW) independently evaluated the methodological quality of each guidelines using AGREE II. Items were scored on a scale from 1 (absence of item) to 7 (item is reported with exceptional quality). Item scores were summed from each reviewer, converted to a percentage, and aggregated to obtain the domain scores. Clinical practice guidelines were ranked as high-quality if 5 or more domains scored >60%, average-quality clinical practice guidelines if 3 or 4 domains scored >60%, and poor quality clinical practice guidelines with two or less domains with scores above 60% [16]. Intra-reviewer scores were tested using a two-way ANOVA with single-rater two-way intra-class correlation coefficients across all guidelines as outlined in a previous study [17]. The degree of reviewer score agreement was defined with the following scale: agreement for intra-class correlation coefficients <0.20, poor; 0.21–0.40, fair; 0.41–0.60, moderate; 0.61–0.80, good; 0.81–1.00, very good [17].

## Comparison of recommendations

We compared the recommendations between all guidelines regardless of their AGREE II score. Two independent co-authors (AH and EW) extracted all recommendations from the included clinical practice guidelines. The final version of the comparative tables of

recommendations were achieved after three rounds of discussion. The sedentary behaviour recommendations were grouped by the following main topics: terminology for sedentary behaviour or time and recommendations. The physical activity recommendations were grouped by the following main topics: aerobic training, resistance training, balance training, and flexibility training. The terminologies and recommendations were compared between the guidelines and summarized descriptively. To determine the certainty of evidence we reviewed the methods and results section of the guidelines; if certainty of evidence was not determined, we reported it as "not reported".

### Ethics

This study does not require ethics approval. No participants were involved in this study.

## Results

### Guideline selection

Our search strategy identified 44 clinical practice guidelines on sedentary behaviour and physical activity. We excluded 26 guidelines as they were not the most recent version of the guideline, were not clinical practice guidelines, or were not available in English (Fig 1). We included 18 guidelines; ten of the 18 guidelines included sedentary behaviour recommendations (S2 Table) and all 18 guidelines include physical activity recommendations (S3 Table).

### Sedentary behaviour/time recommendations

**Type of evidence.**   The sedentary behaviour guidelines used a combination of opinions from stakeholders, other guidelines, systematic reviews, narrative reviews, critical reviews, government reports, and cohort studies to inform the sedentary behaviour guidelines. Brazil 2022 [10], Canada 2020 [12], and USA 2018 [11] used opinions of stakeholders and other guidelines, with Canada 2020 and USA 2018 also utilizing overview of reviews to inform their recommendations. The WHO 2020 [9] used systematic reviews and other guidelines to inform their recommendations, while New Zealand 2013 [18] utilized narrative reviews and two government reports from the USA while Saudi Arabia 2021 [19] utilized other guidelines in individuals <65 years. The UK 2022 [20] used critical reviews and cohort studies to inform their guidelines [9–12,18–21]. Japan 2013 [21], Netherlands 2017 [22], and Qatar 2021 [23] did not specify the type of evidence used to inform their sedentary behaviour guidelines. The WHO 2020 [9], Canada 2020 [12], and Saudi Arabia 2021 [9,12] used the Grading of Recommendations Assessment, Development, and Evaluation (GRADE) approach and New Zealand 2013 [18] used the National Health and Medical Research Council (NHMRC) criteria to assess certainty of evidence. The sedentary behaviour guidelines were developed using very low to low certainty evidence (S2 Table).

**Sedentary behaviour/time definition.**   The Brazil 2022 [10], New Zealand 2013 [18], and UK 2022 [20] guidelines defined sedentary behaviour as activities that require little or no energy expenditure while laying, reclining, or sitting [10,20], while Canada 2020 [12], Netherlands 2017 [22], Qatar 2021 [23], Saudi Arabia 2021 [19], USA 2018 [11], and WHO 2020 [9] defined sedentary behaviour any awake activity include laying, reclining, or siting that uses $\leq$ 1.5 METs (S2 Table). The Japan 2013 [21] guideline defined sedentary behaviour as walking <5,000 steps/day (S2 Table).

**Sedentary behaviour/time recommendations.**   All the sedentary behaviour guidelines suggested reducing total time spent sitting, reallocating prolonged periods of sedentary behaviour to any type of physical activity including light physical activity (e.g., standing), or a

Comparison of national and international sedentary behaviour and physical activity guidelines for older adults: A systematic review and quality appraisal with AGREE II

**Identification**

Studies from databases/registers (n = 13,418)
- OVID Medline (n = 6,736)
- Embase (n = 4726)
- Global Health (n = 971)
- Web of Science (n = 231)
- CINAHL (n = 754)

References from other sources (n = 35)
- Citation searching (n = 1)
- Grey literature (CADTH, SIGN, TRIP, Google) (n =34)

References removed (n = 1,451)
- Duplicates identified manually (n = 0)
- Duplicates identified by Covidence (n = 1,451)
- Marked as ineligible by automation tools (n =0)
- Other reasons (n = 0)

**Screening**

Studies screened (n = 12,002) → Studies excluded (n = 11,958)

Studies sought for retrieval (n = 44) → Studies not retrieved (n = 0)

Studies assessed for eligibility (n = 44) → Studies excluded (n = 26)
- Newer guideline or recommendation (n = 18)
- Study not a guideline or recommendation (n = 7)
- Not available in English (n = 1)

**Included**

Studies included in review (n = 18)

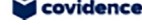

**Fig 1. Identification, screening, eligibility, and included clinical practice guidelines.**

combination of both reducing total time spent sitting and reallocating prolonged sedentary time to physical activity. Three guidelines provided recommendations for limiting specific sedentary behaviours (i.e., screen time). The Canadian 2020 [12] guidelines suggested limiting recreation screen time to <3 hours/day and breaking up long periods of sitting when possible

(very low certainty of evidence). The United Kingdom 2022 [20] guidelines recommended to avoid long periods in screen-based activities (certainty of evidence not reported). The Brazil 2022 [10] guidelines recommend breaking up periods of sedentary behaviour each hour by interspersing at least five minutes of light physical activities such as standing, short walking, and stretching (certainty of evidence not reported).

## Physical activity recommendations

**Type of evidence.** Ten of the 18 guidelines were developed using evidence such as systematic reviews or meta-analyses of randomized controlled trials, with a certainty of evidence rating of not reported [10,18,22,24–26], low [12], moderate to high [9,11], and high [27] (S3 Table). The Canadian 2020 [12], WHO 2020 [9], and the USA 2018 [11] guidelines used the GRADE approach to determine the certainty of evidence. The Australia 2009 [27] and the New Zealand 2013 [18] guideline used the NHMRC criteria, striving to include only level I or II research studies, correlating to systematic reviews and randomized control trials, respectively. The Austrian 2020 [28], Japan 2013 [21], Norway 2001 [29], Poland 2009 [30], and Qatar 2021 [23] guidelines did not specify the type of evidence used to inform their guidelines. China 2021 [31] and Saudi Arabia 2021 [19] utilized other guidelines to inform their decisions on the guidelines, while Denmark 2019 utilized opinions from stakeholders, cohort studies, and randomized controlled trials.

**Physical activity recommendations.** Frequently recommended modes of physical activity included aerobic training, muscle-strengthening/resistance training, balance training, and flexibility training exercises.

*Aerobic exercises.* Sixteen guidelines were consistent in recommending moderate to vigorous intensity aerobic exercise; the Japan 2013 [21] guideline recommends low to moderate intensity by accumulating 40 minutes of 8,000 to 10,000 steps/day while the Qatar 2021 [23] guidelines recommended cardiovascular endurance activities of large muscle groups at least 5 days/week at a moderate intensity or at least 3 days/week at a vigorous intensity. Eight guidelines recommend that older adults accumulate at least 150 minutes of moderate intensity aerobic activity, 75 minutes of vigorous intensity activity, or a combination of both moderate and vigorous intensity each week [9–11,18–20,28,31]. The Norway 2001 [29], India 2012 [24], Australia 2009 [27], and Denmark 2019 [32] guidelines recommend moderate to high intensity aerobic exercise at least 30 minutes/day. The Canada 2020 [12] guidelines recommended achieving 150 minutes of moderate to vigorous intensity aerobic exercise a week, while the Netherlands 2017 [22] recommends at least 150 minutes of moderate aerobic exercise a week. The Germany 2019 [25] guidelines recommend moderate aerobic exercise >5 days/week, with the total aerobic exercise time exceeding 150 minutes. The Poland 2009 [30] guidelines recommend moderate exercise at least 3 days/week, for an average of 40 minutes/day.

*Strength training.* Fifteen guidelines provided recommendations for strength training [9–12,18–20,22–25,27,28,30,31] but only five of the 15 guidelines recommended strength training at a moderate to high intensity [9–11,23,27]. Seven guidelines recommended older adults engage in strength training of major muscle groups 2 days/week [11,12,19,22,24,25,31] while seven guidelines recommended to train ≥ 2 days/week [10,18,20,23,28]. The Poland 2009 [30] guidelines did not provide a recommended frequency. The Australia 2009 [27] guidelines recommend 2 to 3 days/week of resistance training with the added detail to focus on upper (biceps, shoulder flexion, chest press, back row) and lower body (hamstrings, quadriceps, leg press, calves) areas with 2 to 3 sets, and 10 to 12 repetitions in each set. The WHO 2020 [9] guidelines recommend 3 days/week of moderate or greater muscle training. The Poland 2009 [30] guidelines recommend resistance training to be at least 10% to 15% of the overall

exercising routine. The Qatar 2021[23] guidelines suggested 8–10 compound exercises targeting major muscle groups at least $\geq$ 2 days/week with 48 hours rest for same muscle groups at light (40–50% 1 RM) or moderate (60–70% 1 RM) intensity. Strength training exercises should be performed at 8–12 reps/set with 1–2 sets of each exercise at a moderate speed (6 sec/rep), 2–3 min rest between sets [23]

*Balance training.* Eight guidelines provided recommendations for balance training (S3 Table). The WHO 2020 [9] guidelines recommend 3 days/week of moderate challenging balance training. The New Zealand 2013 [18] guidelines recommend 3 days/week of balance training with no guidance on intensity or time, while the Netherlands 2017 [22], Saudi Arabia 2021 [19], and UK 2022 [20] guidelines recommend at least 2 days/week of balance training with no guidance on intensity or time. The Australia 2009 [27] guidelines recommend 1 to 7 days/week of balance training focused on dynamic mobility, static, and one-leg stance.

*Flexibility training.* Six guidelines reported on flexibility training (S3 Table). The WHO 2020 [9] guidelines recommend 3 days/week of moderate or greater flexibility training. The New Zealand 2013 [18] guidelines recommend 3 days/week of flexibility training. The Australia 2009 [27] guidelines recommend 2 to 3 days/week of moderate flexibility training for 10 to 30 seconds for each stretch, repeating 3 to 4 times. The UK 2022 [20] guidelines recommend at least 2 days/week of flexibility training. The Saudi Arabia 2021 [19] guidelines provided a strong recommendation of 2 days/week of flexibility training, while the the Qatar 2021 [23] guidelines suggest stretching major muscle groups 2–3 days/week, where each stretch should be held for 10–30 sec to the point of tightness or slight discomfort.

## Considerations for diverse populations and modifications for co-morbidities

There was one guideline that provided recommendations using cultural accommodations. In April 2023, the Canadian Guidelines provided a culturally-focused update to the 24-Hour Movement Guidelines for the Punjabi older adult population [33]. The research team employed a member of the Punjabi community and research advisor to interview members of the community and bring their knowledge to the steering committee [33]. Seven guidelines proposed considerations for physical activity guidelines for people with co-morbidities. The New Zealand 2013 [18] and Australia 2009 [27] guidelines recommended for older adults with a health condition or co-morbidities to gradually build up to the recommended daily physical activity levels with the assumption that gradually increasing physical activity levels may prevent injuries [18]. The New Zealand guidelines also recommended aerobic activities can be more challenging for older adults who are frail, and so a good option is low-intensity resistance activities combined with some aerobic activity, such as repeated sit-to-stand exercises [18]. The India 2012 [24] guidelines also recommended that sudden initiation of acceleration in movements should be avoided in older adults, especially those with co-morbidities such as congestive heart failure. The USA 2018 [11] and Brazil 2022 [10] guidelines suggest that older adults with chronic conditions should understand whether and how their conditions affect their ability to conduct regular physical activity safely. The USA 2018 [11], Austria 2020 [28], and Brazil 2022 [10] guidelines recommend for older adults with chronic conditions to be as physically active as their conditions allow for. The WHO 2020 [9] guidelines recommend that older adults with chronic conditions should attempt to complete varied multicomponent physical activity at moderate or high intensity at least 3 days/week in order to enhance functional capacity and prevent falls.

The UK 2022 [20] guideline addressed sedentary behaviour in frail older adults. The guideline recommends that any increase in volume and frequency of light activities, and reduction

in sedentary time, would be a starting point [20]. As strenuous activities are less feasible, a programme of activities can focus on engaging in regular sit-to-stand exercise and short walks, stair climbing, embedding strength and balance activities into everyday life tasks, and increasing the duration of walking, rather than concentrating on intensity [20].

## Quality assessment

Our inter-rater reliability was mostly good to very good (0.60–1.00) (Table 1). Overall, the guidelines were strongest in domain 4 (Clarity of Presentation) and domain 6 (Editorial Independence) (Table 1).

The guidelines published by the World Health Organization, USA, Germany, Canada, and Australia met the criteria for high quality, scoring at least 60% in at least 5 domains (Table 2). Guidelines published by Brazil, China, Netherlands, New Zealand, Norway, and United Kingdom met the criteria for average quality, scoring at least 60% in three to four domains. Guidelines published by Australia and New Zealand, Denmark, South India, Japan, Poland, and India had two or less domains that scored over 60% and were considered low quality.

## Discussion

The purpose of our study was to compare the international sedentary behaviour and physical activity guidelines. We identified ten sedentary behaviour guidelines and eighteen physical activity guidelines. We found that all the sedentary behaviour guidelines were developed using high risk of bias evidence and very low to low certainty of evidence. The terminology to define sedentary behaviour and time were inconsistent between guidelines. All the sedentary behaviour guidelines suggested reducing total time spent sitting, reallocating prolonged periods of sedentary behaviour to any type of physical activity including light physical activity (e.g.,

**Table 1. Summary of AGREE II and intraclass correlation coefficient (ICC).**

| Guideline | ICC | Degree of Agreement |
|---|---|---|
| Australia 2009 [27] | 0.69 | Good |
| Austrian 2020* [28] | NA | |
| Brazil 2022 [10] | 0.86 | Very Good |
| Canadian 2020 [12] | 0.66 | Good |
| Chinese 2021 [31] | 0.86 | Very Good |
| Denmark 2019 [32] | 0.93 | Very Good |
| German 2019 [25] | 0.71 | Good |
| India 2012 [24] | 0.71 | Good |
| Japanese 2013 [21] | 0.85 | Very Good |
| Netherlands 2018 [22] | 0.91 | Very Good |
| New Zealand 2013 [18] | 0.85 | Very Good |
| Norway 2017 [29] | 0.78 | Good |
| Polish 2009 [30] | 0.89 | Very Good |
| Qatar 2021 [23] | 0.86 | Very Good |
| Saudi Arabia 2021 [19] | 0.79 | Good |
| UK 2021 [20] | 0.47 | Moderate |
| USA 2018 [11] | 0.89 | Very Good |
| WHO 2020 [9] | 0.72 | Good |

ICC <0.20, poor; 0.21–0.40, fair; 0.41–0.60, moderate; 0.61–0.80, good; 0.81–1.00, very good.

*The full Austrian 2020 guideline was not available in English. Domains >60 are considered high-quality.

**Table 2. AGREE II scores.**

| Guideline | Domain | | | | | |
|---|---|---|---|---|---|---|
| | Scope & Purpose | Stakeholder Involvement | Rigour of Development | Clarity of Presentation | Applicability | Editorial Independence |
| Australia 2009 [27] | 75% | 83% | 85% | 89% | 38% | 50% |
| Austrian 2020* [28] | | | | | | |
| Brazil 2022 [10] | 67% | 50% | 63% | 72% | 48% | 100% |
| Canadian 2020 [12] | 72% | 92% | 92% | 89% | 94% | 100% |
| Chinese 2021 [31] | 44% | 81% | 32% | 67% | 58% | 75% |
| Denmark 2019 [32] | 56% | 36% | 7% | 72% | 4% | 71% |
| German 2019 [25] | 89% | 67% | 61% | 78% | 65% | 100% |
| India 2012 [24] | 94% | 72% | 74% | 89% | 65% | 25% |
| Japanese 2013 [21] | 47% | 44% | 53% | 42% | 42% | 92% |
| Netherlands 2018 [22] | 47% | 19% | 25% | 50% | 6% | 58% |
| New Zealand 2013 [18] | 78% | 42% | 70% | 67% | 10% | 79% |
| Norway 2017 [29] | 92% | 86% | 59% | 89% | 63% | 33% |
| Polish 2009 [30] | 56% | 33% | 33% | 56% | 27% | 0% |
| Qatar 2021 [23] | 56% | 36% | 22% | 97% | 56% | 63% |
| Saudi Arabia 2021 [19] | 58% | 67% | 61% | 97% | 54% | 96% |
| UK 2021 [20] | 81% | 58% | 64% | 86% | 29% | 100% |
| USA 2018 [11] | 97% | 97% | 85% | 97% | 83% | 100% |
| WHO 2020 [9] | 94% | 72% | 74% | 89% | 65% | 25% |
| **Mean Score** | **67%±20** | **58%±25** | **52%±27** | **75%±20** | **44%±28** | **67%±34** |

*The full Austrian 2020 guideline was not available in English. Domains >60 are considered high-quality.

standing), or a combination of both reducing total time spent sitting and reallocating prolonged sedentary time to physical activity. The physical activity guidelines were based on higher quality and certainty evidence. The recommendations for aerobic and resistance training were consistent between guidelines, which recommended accumulating 150 minutes per week of moderate to vigorous aerobic exercise, 2 days/week of strength; the evidence for aerobic and resistance training is based on moderate to high quality and certainty evidence. Few guidelines provided recommendations on balance training, but all guidelines were consistent in recommending at least 3 days/week of balance training. The evidence used to develop the balance training recommendations were based on moderate quality evidence. Only six studies provided recommendations on flexibility training; it is unclear what type of evidence was used to develop the flexibility training recommendations.

The evidence used to develop the recommendations for the sedentary behaviour guidelines was poor, while the evidence to develop of the physical activity guidelines was moderate-to-high. From our review, we found most physical activity guidelines were developed using systematic reviews and meta-analysis of randomized controlled trials, while the sedentary behaviour guidelines were mainly developed using stakeholder's opinions and other guidelines; only the Canadian 2021 [12], USA 2018 [11], and WHO 2020 [9] guidelines utilized additional systematic reviews of randomized controlled trials to inform their recommendations. The British Medical Journal New Evidence Pyramid suggests that a hierarchy of evidence exists such that randomized controlled trials are ranked above cohort studies; however, internal validity (i.e., risk of bias) may alter the hierarchy such that a high risk of bias randomized controlled trial may be ranked lower than a high-quality cohort study [34]. In our review, we found that all studies used to develop the sedentary behaviour recommendations were considered high risk of bias; the most used tool to assess risk of bias was the Cochran risk of bias tool. On the other

hand, the aerobic, resistance, and balance training recommendations were based on low risk of bias. In addition, we evaluated if the guidelines used either the GRADE [35] or NHMRC [36] approaches, which are systematic methods that involve assessing the quality of evidence and strength of the recommendations. We found two guidelines used the GRADE approach to develop the sedentary behaviour guidelines [9,12], while five guidelines used GRADE or NHMRC to develop the physical activity guidelines [9,11,12,27]. In the WHO 2020 guidelines, replacing sedentary behaviour with physical activity of any intensity was a recommendation; however, the certainty of evidence is low [9]. In the Canada 2020 guidelines, recommendations were made to limit sedentary time to <8 hours/day, with less than three hours of recreational screen time, and to break up long periods of sitting; however, the certainty of evidence for the recommendation is also very low [12]. According to the GRADE and NHMRC approaches, low certainty evidence indicates that further research and that further research is likely to change the estimated results [37]. On the other hand, we found that the recommendations for the aerobic, resistance, and balance training recommendations were based on moderate to high certainty evidence. Thus, the current research used to develop the sedentary behaviour guidelines does not provide a suitable foundation to implement guidelines into practice. Future research, taking into consideration approaches such as risk of bias and certainty of evidence are necessary to define sedentary behaviour or time and then develop suitable interventions.

There is a debate about how to target sedentary time and behaviour among older adults. Previous studies focused on reducing total sedentary time [38,39], while other studies aimed to increase physical activity levels with the assumption that sedentary time will be reallocated to physical activity [40,41]. Six guidelines recommended older adults decrease time spent in sedentary behaviours [9–12,18,21,22], while three guidelines recommended to reallocate sedentary time to time spent in light physical activity [9,10,20]. A 2021 Cochrane review synthesized the results from seven studies (six randomized controlled trials and one cluster-randomized controlled trial) that targeted interventions to reduce total sedentary time among community dwelling older adults; the interventions focused on strategies like counselling, goal setting, and information sessions to reduce sedentary time and behaviour [13]. The authors of the Cochrane review concluded that it is not clear what interventions are effective at reducing total sedentary time in older adults (mean 44.91 minutes/day lower, 95% confidence intervals [CI] 93.13 minutes/day lower to 3.32 minutes/day higher, low certainty of evidence, 7 studies, 397 participants) [13]. Some of the sedentary behaviour guidelines recommended allocating sedentary time to aerobic activity including light aerobic training; however, as indicated in physical activity guidelines [9,11,12,27], resistance training is important for bones and muscles. A possible solution to decreasing sedentary behaviour or reducing sedentary time is to embed activities of daily living into one's routine. In the Lifestyle integrated Functional Exercise (LiFE) study, the intervention group embedded functional strength and balance training in daily activities (e.g., while a kettle is boiling, try balancing on one foot or use the counter to do push-ups) [42]. Researchers reported a significant reduction of 31% in the rate of falls for the LiFE programme group compared with controls [42]. We can extrapolate evidence from the LiFE study into a possible intervention to decrease sedentary behaviours, where older adults focus on small, attenable changes made to daily activities that increase the energy load of these activities. Further work should be done to solidify actionable steps to address sedentary behaviour in older adults, whether it is to introduce physical activity, focus on reducing sedentary time itself, or through novel concepts.

From a diversity and inclusion perspective, we found one guideline focused on Punjabi cultural accommodations when developing recommendations. There exists the need for increased intersectional considerations when addressing sedentary behaviour and physical activity

guidelines in older adults. A cross-sectional study identified older, retired individuals with less education and lower incomes as a having higher odds of spending less time in leisure-time physical activity and being at risk of living a sedentary lifestyle [6]. The cross-sectional study indicates that those of lower socioeconomic status may have reduced access to knowledge and activities that reduce sedentary behaviour. A qualitative study identified seniors centres and places of worship as critical resources in the community [43]. These publicly funded resources encouraged engagement in life activities and older adults to remain active [43]. To improve uptake and adoption of guidelines, it is important to integrate cultural, educational, and socio-economic factors when developing recommendations. Clinical practice guidelines should address sedentary behaviour and physical activity through a holistic lens. A possible solution would be to include diverse individuals on the research team, and to utilize patient partners of diverse backgrounds in development of guidelines. The 2023 Punjabi focused update of the Canadian Guidelines utilized a research member of the culture to facilitate the tailoring of the materials to their culture [33]. Patient representation on research teams have improved the conduct of guideline development, scope, inclusion of patient-relevant topics, outcome selection, and planned approaches to the guideline development [44]. To facilitate health care providers in providing applicable care for older individuals of all socioeconomic levels and cultural backgrounds, future sedentary behaviour research could investigate interventions with a focus on diversity, and inclusion.

## Limitations and strengths

The methodological quality of our study is in line with previous work; however, it is not without its limitations. One limitation was that the search strategy was conducted in English, which may have filtered guidelines from other countries. In addition, while the AGREE II tool assesses the methodological quality it does not assess clinical content. Our study also had several strengths. Strengths of the study includes the use of the AGREE II tool in methodological assessment of guidelines. This is a validated tool and used by experts previously to evaluate clinical practice guidelines. Another strength is that this manuscript provides a foundation for upcoming projects to address key knowledge gaps related to sedentary behaviour research.

## Conclusion

We identified 18 international guidelines for older adults; ten guidelines provided recommendations for sedentary behaviour and 18 for physical activity. We found that sedentary behaviour guidelines were generally developed with lower quality evidence using cohort studies and knowledge user opinions, while the physical activity guidelines were developed using moderate quality evidence from overview of reviews and systematic reviews/meta-analysis of randomized controlled trials. The definition and recommendations for sedentary behaviour or time was not consistent across guidelines. The aerobic and resistance training recommendations were consistent between guidelines and developed using moderate to high quality and certainty evidence. Only eight guidelines provided recommendations for balance training, but the evidence was based on moderate quality and certainty evidence. Only six guidelines provided evidence on flexibility training which was based on low quality and certainty evidence.

## Supporting information

**S1 Checklist.**
(PDF)

**S1 Table. Search strategy.**
(PDF)

**S2 Table. Summary of the sedentary behaviour guidelines.**
(DOCX)

**S3 Table. Summary of the physical activity guidelines.**
(DOCX)

**S4 Table. PRISMA 2020 checklist.**
(PDF)

## Acknowledgments

The authors would like to thank the following funding agents for their support with the project including McMaster Institute for Research on Aging (MIRA), AGE-WELL, and the Hamilton Health Sciences New Investigator Fund. We would like to acknowledge Charlotte Beck, Librarian at the University of British Columbia, for her assistance in designing the search strategy.

### Other information

Amy Huang is a trainee who helped lead this review under the supervision of Dr. Isabel B. Rodrigues. Ellen Wang, Stephanie Sanger, and Dr. Alexandra Papaioannou are collaborators on the project. The authors have no competing interests. The principal investigator (IBR) received funding support from the Hamilton Health Sciences New Investigator Fund, AGE-WELL-McMaster Institute for Research on Aging (MIRA) Award. The funders had no role in study design, data collection and analysis, decision to publish, or preparation of the manuscript. IBR received a salary from the AGE-WELL-McMaster Institute for Research on Aging (MIRA) Award. This review was retrospectively registered on Prospero (https://www.crd.york.ac.uk/prospero/) ID 425682. There is no protocol available, and no amendment were made to the protocol. Data is publicly available upon request for research purposes only.

## Author Contributions

**Conceptualization:** Isabel B. Rodrigues.

**Data curation:** Amy Huang, Ellen Wang, Stephanie Sanger.

**Formal analysis:** Amy Huang.

**Funding acquisition:** Isabel B. Rodrigues.

**Methodology:** Alexandra Papaioannou, Isabel B. Rodrigues.

**Project administration:** Isabel B. Rodrigues.

**Resources:** Ellen Wang, Isabel B. Rodrigues.

**Supervision:** Isabel B. Rodrigues.

**Writing – original draft:** Amy Huang, Isabel B. Rodrigues.

**Writing – review & editing:** Ellen Wang, Alexandra Papaioannou, Isabel B. Rodrigues.

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
