## [Decision Letter · Decision Letter 0]

15 Aug 2023

PONE-D-23-13100Comparison of national and international sedentary behaviour and physical activity guidelines for older adults: A systematic review and quality appraisal with AGREE IIPLOS ONE

Dear Dr. Rodrigues,

Thank you for submitting your manuscript to PLOS ONE. After careful consideration, we feel that it has merit but does not fully meet PLOS ONE’s publication criteria as it currently stands. Therefore, we invite you to submit a revised version of the manuscript that addresses the points raised during the review process.

We look forward to receiving your revised manuscript.

Kind regards,

Stevo Popovic, Ph.D.

Academic Editor

PLOS ONE

Journal Requirements:

5. We note that this manuscript is a systematic review or meta-analysis; our author guidelines therefore require that you use PRISMA guidance to help improve reporting quality of this type of study. Please upload copies of the completed PRISMA checklist as Supporting Information with a file name “PRISMA checklist”.

Additional Editor Comments:

Dear Author(s),

I am so please to inform you I have collected three reviews and we are ready to go ahead with the evaluation process. It is your turn now to read the reviews and carefully revise the manuscript according to the requirements of the reviewers. I have to highlight that there are two positive reviews and one negative, so I would appreciate if you prepare the revision of your manuscript in line of two positive reviews but also to consider to negative one and accept all the comments you believe that can help improving the manuscript; however, I would appreciate if you prepare the comments with some arguments and adequate justifications for the rest and try to persuade the reviewer to change his mind.

Sincerely.

Reviewers' comments:

Reviewer's Responses to Questions

**Comments to the Author**

1. Is the manuscript technically sound, and do the data support the conclusions?

Reviewer #1: Yes

Reviewer #2: Yes

Reviewer #3: Yes

2. Has the statistical analysis been performed appropriately and rigorously? 

Reviewer #1: Yes

Reviewer #2: Yes

Reviewer #3: Yes

3. Have the authors made all data underlying the findings in their manuscript fully available?

Reviewer #1: Yes

Reviewer #2: Yes

Reviewer #3: Yes

4. Is the manuscript presented in an intelligible fashion and written in standard English?

Reviewer #1: Yes

Reviewer #2: Yes

Reviewer #3: Yes

5. Review Comments to the Author

Reviewer #1: Lines 71-71 - this sentence requires a quotation

Lines 93-99 - Table S1 shows a fuzzy index database search! Why was the VoS database not used? A combination of keywords should be displayed for searching manuscripts

Lines 143-145 - These data do not agree with those in Figure 1

Lines 257-262 - this text should be placed between Table 1 and Table 2

Lines 381-383 - this has already been discussed, it should not be repeated. Here, conclusions should be given according to the author's opinion

Reviewer #2: I notice a lot of ambiguities: 1) Why does systematic review and quality assessment work at the same time? isn't that much on thay plate?; 2) I don't see a clear idea, but several ideas that bring confusion. It is difficult to follow the flow of research work. It is true that insight into best practices helps avoid duplication, and identify knowledge gaps. The authors mention it in Line 77, but later they have a completely different goal, and that goal is not very clear.; 3) Why were Web of Science and Scopus not searched?; 5) Why is a partial English copy of guidelines used?...

The complete method is such that the research cannot be repeated. The author's obligation is to make it possible to repeat the same procedure.

The structure is adequate. There are all the elements that the article should have. But the chapters themselves need to be improved. And the question is whether the idea is correct. But this manuscript does not have the strength required for a journal of Plos One quality.

Reviewer #3: First of all, I can say that the authors have chosen a good topic, and the problem that was the subject of analysis in the following years will be more and more relevant, given that the average human life is longer and that more and more people are in the so-called third age. In that sense, the work has potential for readers. the work is methodologically well laid out, from the abstract to each individual part of the work. In the introduction.The work method is adequate, and as the authors themselves admit in the part of the limitations of the study, more relevant results would certainly have been obtained if the study sample had been larger and better, if recommendations from other speaking areas besides English had been included. Regardless, the results of the research, as well as the discussion that was done with quality, directed the authors to accurate and explicit conclusions. The references used are adequately listed with a fair number of younger ones. In any case, I recommend that the work be accepted.n, the key terms related to the issue are well explained, and they are clearly supported by appropriate references.

6. PLOS authors have the option to publish the peer review history of their article (what does this mean?). If published, this will include your full peer review and any attached files.

Reviewer #1: **Yes: **Jovan Gardasevic, University of Montenegro

Reviewer #2: No

Reviewer #3: No

---

## [Author Response · Author response to Decision Letter 0]

6 Oct 2023

Thank you for reviewing and providing feedback on our manuscript "Comparison of national and international sedentary behaviour and physical activity guidelines for older adults: A systematic review and quality appraisal with AGREE II". Revisions are noted as tracked changes within the manuscript and indicated below under each reviewer’s comments. The corresponding page and line numbers are associated with the track changed manuscript.

Reviewer 1:

1. Lines 71-71 - this sentence requires a quotation.

We have included quotes for the following: Sedentary behaviour is defined as “any waking behaviour characterized by an energy expenditure of 1.5 metabolic equivalents (METs) or lower while sitting, reclining, or lying”, while sedentary time is measured by “the amount of time spent in these positions” (page 2, lines 49 to 51). 

2. Lines 93-99 - Table S1 shows a fuzzy index database search! Why was the VoS database not used? A combination of keywords should be displayed for searching manuscripts

We consulted with two health science librarians from McMaster University and the University of British Columbia to review our search strategy; Web of Science does not have guidelines as a publication type but rather this database includes articles that discuss guidelines. Nevertheless, to address your point, we conducted another search in the Web of Science and CINAHL databases. We identified 231 studies in Web of Science, and 754 in CINAHL. We also conducted additional searches in Canada’s Drug and health Technology Agency Grey Matters Database and the Scottish Intercollegiate Guideline Network. Through our new search, we identified six additional guidelines; we included two of the six guidelines and excluded the other four as they were not the latest version of the guideline. 

We included the following statement in our manuscript: “On August 22nd, 2023, we expanded our search to include Web of Science and CINAHL. We also searched for additional guidelines through Canada’s Drug and health Technology Agency Grey Matters Database (https://www.cadth.ca/) and the Scottish Intercollegiate Guideline Network (https://www.sign.ac.uk/). A second librarian the University of British Columbia reviewed the search strategy.” (page 4, lines 105 to 109). 

3. Lines 143-145 - These data do not agree with those in Figure 1

We updated Figure 1 and the manuscript, so it now reads: “Our search strategy identified 44 clinical practice guidelines on sedentary behaviour and physical activity. We excluded 26 guidelines as they were not the most recent version of the guideline, were not clinical practice guidelines, or were not available in English (Fig 1). We included 18 guidelines; ten of the 18 guidelines included sedentary behaviour recommendations (S2 Table) and all 18 guidelines include physical activity recommendations (S3 Table).” (page 6, lines 154 to 159).

4. Lines 257-262 - this text should be placed between Table 1 and Table 2

The following text was moved between Table 1 and Table 2: “The guidelines published by the World Health Organization, USA, Germany, Canada, and Australia met the criteria for high quality, scoring at least 60% in at least 5 domains (Table 2). Guidelines published by Brazil, China, Netherlands, New Zealand, Norway, and United Kingdom met the criteria for average quality, scoring at least 60% in three to four domains. Guidelines published by Australia and New Zealand, Denmark, South India, Japan, Poland, and India had two or less domains that scored over 60% and were considered low quality.” (page 12, lines 312 to 317).

5. Lines 381-383 - this has already been discussed, it should not be repeated. Here, conclusions should be given according to the author's opinion

We have removed the following text: “Regarding the language limitation on the search strategy, we have exhausted all options in finding translated versions of guidelines. We were able to obtain a partial English copy of the Japanese guidelines.”

Reviewer 2: I notice a lot of ambiguities: 

1. Why does systematic review and quality assessment work at the same time? isn't that much on thay plate?

We agree that incorporating both a systematic review and quality assessment into a single manuscript is a significant undertaking; however, it is typical to conduct a systematic review and quality assessment at the same time. We followed the guidelines recommended by AGREE II, which was specifically designed to evaluate the methodological quality of practice guidelines during the review process (https://www.agreetrust.org/resource-centre/agree-ii/agree-ii-instructions/). Furthermore, previous PLOS One publication including "Methodological quality of clinical practice guidelines with physical activity recommendations for people diagnosed with cancer: A systematic critical appraisal using the AGREE II tool" (

https://doi.org/10.1371/journal.pone.0214846) and “Assessing methodological quality of Russian clinical practice guidelines and introducing AGREE II instrument in Russia (https://doi.org/10.1371/journal.pone.0203328) have conducted systematic reviews and quality assessments at the same time. 

2. I don't see a clear idea, but several ideas that bring confusion. It is difficult to follow the flow of research work. It is true that insight into best practices helps avoid duplication, and identify knowledge gaps. The authors mention it in Line 77, but later they have a completely different goal, and that goal is not very clear. 

We have revised the wording to ensure clarity throughout the manuscript. Our updated statement is as follows: “In the past decade, numerous countries have reviewed or updated their sedentary behaviour guidelines for older adults, with a trend toward more evidence-based guidelines [9–12]. A synthesis of international sedentary behaviour guidelines can help reveal more information about interventions that may be used to decrease sedentary behaviours, time spent engaging in these behaviours, and methods to decrease total sedentary time in older adults. In addition, most interventions have attempted to decrease sedentary time by increasing physical activity levels with the assumption that sedentary time would be reallocated to physical activity [13]. Therefore, reviewing physical activity guidelines can also provide insight into effective interventions that target sedentary behaviour since most sedentary behaviour guidelines are part of the physical activity guidelines [9–12]. Lastly, comparing sedentary behaviour and physical activity guidelines can provide insight into best practices, avoid duplication, and identify knowledge gaps. The purpose of this systematic review was to separately survey available sedentary behaviour and physical activity guidelines from around the world to compare recommendations and critically analyze the methodology through which the guidelines were developed.” (page 3, line 71 to 91). 

3. Why were Web of Science and Scopus not searched?

When we initially consulted with the librarian on our team, we chose not to use Web of Science or Scopus because these are not subject databases. Web of Science and Scopus do not have guidelines as a publication type, but rather, these database includes articles that discuss guidelines. We did not conduct a search in Scopus since the database publishes “primary document types from serial publications. Primary means that the author is identical to the research in charge of the presented findings”( https://www.elsevier.com/solutions/scopus/how-scopus- works/content). Guidelines are not a primary document. However, since Reviewer 1 also suggested we conduct our search in Web of Science we have updated our search strategy. This was our response to Reviewer 1: 

“We consulted with two health science librarians from McMaster University and the University of British Columbia to review our search strategy; Web of Science does not have guidelines as a publication type but rather this database includes articles that discuss guidelines. Nevertheless, to address your point, we conducted another search in the Web of Science and CINAHL databases. We identified 231 studies in Web of Science, and 754 in CINAHL. We also conducted additional searches in Canada’s Drug and health Technology Agency Grey Matters Database and the Scottish Intercollegiate Guideline Network. Through our new search, we identified six additional guidelines; we included two of the six guidelines and excluded the other four as they were not the latest version of the guideline. 

We included the following statement in our manuscript: “On August 22nd, 2023, we expanded our search to include Web of Science and CINAHL. We also searched for additional guidelines through Canada’s Drug and health Technology Agency Grey Matters Database (https://www.cadth.ca/) and the Scottish Intercollegiate Guideline Network (https://www.sign.ac.uk/). A second librarian the University of British Columbia reviewed the search strategy.” (page 4, lines 105 to 109).” 

5. Why is a partial English copy of guidelines used?...

The full version of the Austrian guideline was not available in English. We were only able to locate a one-page document of the Austrian guideline that was written in English; the one page document was translated by the authors of the guideline. Thus, we were not able to critically appraise the quality of the guideline, but we were able to include the physical activity recommendations as they were translated to English. To address your concern, we have the following sentence: “The full Austrian 2020 guideline was not available in English” (page 12, line 309 to 310 and page 13, line 319 to 320). 

The complete method is such that the research cannot be repeated. The author's obligation is to make it possible to repeat the same procedure. The structure is adequate. There are all the elements that the article should have. But the chapters themselves need to be improved. And the question is whether the idea is correct. But this manuscript does not have the strength required for a journal of Plos One quality.

We sincerely thank the reviewer for their insightful comments. We have carefully incorporated the reviewer's valuable feedback into our manuscript revisions and conducted a new search in Web of Science and CINHAL. Our librarian also helped search for additional guidelines in CDHT (Canada’s Drug and Health Technology Agency Grey Matters Database) and SIGN (Scottish Intercollegiate Guideline Network). Through our search we identified two additional guidelines. We hope these changes align with Reviewer 2’s expectations and kindly invite them to reconsider their assessment.

Reviewer 3: First of all, I can say that the authors have chosen a good topic, and the problem that was the subject of analysis in the following years will be more and more relevant, given that the average human life is longer and that more and more people are in the so-called third age. In that sense, the work has potential for readers. the work is methodologically well laid out, from the abstract to each individual part of the work. In the introduction. The work method is adequate, and as the authors themselves admit in the part of the limitations of the study, more relevant results would certainly have been obtained if the study sample had been larger and better, if recommendations from other speaking areas besides English had been included. Regardless, the results of the research, as well as the discussion that was done with quality, directed the authors to accurate and explicit conclusions. The references used are adequately listed with a fair number of younger ones. In any case, I recommend that the work be accepted.n, the key terms related to the issue are well explained, and they are clearly supported by appropriate references.

Thank you for taking the time to read our manuscript and provide feedback. Your recognition of our chosen topic's relevance, methodological soundness, and quality of discussion is deeply appreciated. Your feedback encourages us to refine our work further. We are grateful for your recommendation and look forward to contributing meaningfully to the field.

Academic Editor: In addition to a point-by-point response to the comments from Reviewers #1, #2, and #3, I will highlight a few comments that are important to address. General Comments: When submitting your revision, we need you to address these additional requirements.

We have changed the file name of our document to Main body as outlined in the https://journals.plos.org/plosone/s/file?id=wjVg/PLOSOne_formatting_sample_main_body.pdf example. 

We have also changed the order of author affiliations to Department, Institution, City, State, Country as outlined in https://journals.plos.org/plosone/s/file?id=ba62/PLOSOne_formatting_sample_title_authors_affiliations.pdf. 

We have provided the correct information for the grants in the ‘Funding Information’ and ‘Financial Disclosure’ sections.

3. Thank you for stating the following financial disclosure: "The funders had no role in study design, data collection and analysis, decision to publish, or preparation of the manuscript." At this time, please address the following queries:

We received financial support from the Hamilton Health Sciences New Investigators Fund, AGE-WELL-McMaster Institute for Research on Aging (MIRA) Award (page 22, lines 608 to 610). 

We have included the following sentence in the manuscript: “The funders had no role in study design, data collection and analysis, decision to publish, or preparation of the manuscript.” (page 22, lines 610 to 611). 

We have included the following sentence in the manuscript: “IBR received a salary from the AGE-WELL-McMaster Institute for Research on Aging (MIRA) Award” (page 22, lines 611 to 613). 

N/A

N/A

As this is a systematic review, there is no de-identified data set. All of the data that is necessary to replicate our study findings are available in the tables and figures provided. We are referring to:

S1 Table. Search strategy. 

S2 Table. Summary of the sedentary behaviour guidelines.

S3 Table. Summary of the physical activity guidelines.

S4 Table. PRISMA 2020 Checklist.

Fig 1. Identification, screening, eligibility, and included clinical practice guidelines. 

5. We note that this manuscript is a systematic review or meta-analysis; our author guidelines therefore require that you use PRISMA guidance to help improve reporting quality of this type of study. Please upload copies of the completed PRISMA checklist as Supporting Information with a file name “PRISMA checklist”.

We have updated the document S4 Table. PRISMA 2020 Checklist to account for the changes in the clean version (i.e., Main body_clean).

---

## [Decision Letter · Decision Letter 1]

9 Nov 2023

Comparison of national and international sedentary behaviour and physical activity guidelines for older adults: A systematic review and quality appraisal with AGREE II

PONE-D-23-13100R1

Dear Dr. Rodrigues,

We’re pleased to inform you that your manuscript has been judged scientifically suitable for publication and will be formally accepted for publication once it meets all outstanding technical requirements.

Kind regards,

Stevo Popovic, Ph.D.

Academic Editor

PLOS ONE

Additional Editor Comments (optional):

Dear Author(s),

I am so please to inform you I have collected two positive reviews from principal reviewers and we are ready to complete this review process. Therefore, I would like to thank you for hard work and inform you that your manuscript is provisionally accepted and send to the journal office for the further evaluation process .

Sincerely,

Handling Editor

Reviewers' comments:

Reviewer's Responses to Questions

**Comments to the Author**

1. If the authors have adequately addressed your comments raised in a previous round of review and you feel that this manuscript is now acceptable for publication, you may indicate that here to bypass the “Comments to the Author” section, enter your conflict of interest statement in the “Confidential to Editor” section, and submit your "Accept" recommendation.

Reviewer #1: (No Response)

Reviewer #3: All comments have been addressed

2. Is the manuscript technically sound, and do the data support the conclusions?

Reviewer #1: (No Response)

Reviewer #3: Yes

3. Has the statistical analysis been performed appropriately and rigorously? 

Reviewer #1: (No Response)

Reviewer #3: Yes

4. Have the authors made all data underlying the findings in their manuscript fully available?

Reviewer #1: (No Response)

Reviewer #3: Yes

5. Is the manuscript presented in an intelligible fashion and written in standard English?

Reviewer #1: (No Response)

Reviewer #3: Yes

6. Review Comments to the Author

Reviewer #1: (No Response)

Reviewer #3: As far as I can see, all the doubts about the work mentioned by the reviewers have been successfully resolved, and I believe that the work can be accepted for publication!

7. PLOS authors have the option to publish the peer review history of their article (what does this mean?). If published, this will include your full peer review and any attached files.

Reviewer #1: **Yes: **Jovan Gardasevic

Reviewer #3: No

---

## [Editor Report · Acceptance letter]

13 Nov 2023

PONE-D-23-13100R1 

Comparison of national and international sedentary behaviour and physical activity guidelines for older adults: A systematic review and quality appraisal with AGREE II 

Dear Dr. Rodrigues:

I'm pleased to inform you that your manuscript has been deemed suitable for publication in PLOS ONE. Congratulations! Your manuscript is now with our production department. 

Kind regards, 

on behalf of

Professor Stevo Popovic 

Academic Editor

PLOS ONE